# The Teaching–Regret–Stability Principle in Non-Stationary Reinforcement Learning

## Abstract

Standard treatments of non-stationary reinforcement learning primarily emphasize tracking and evaluate performance via dynamic regret under variation-budget drift. In many deployments, however, practitioners may also care about *which* policy is learned (e.g., compliance/safety targets) and how smoothly it evolves over time. This motivates studying teaching to a target policy and policy-trajectory stability alongside regret, as complementary objectives rather than replacements. We formalize this viewpoint in the *Teaching–Regret–Stability (TRS) Principle* for *Teachable Non-stationary RL (TNRL)*. Under standard variation-budget assumptions and a Lipschitz policy-update condition, we prove a high-level theorem showing that a bounded-budget teacher can simultaneously drive the teaching error to an arbitrarily small target, keep dynamic regret sublinear, and ensure that the policy sequence remains stable on average.

## 1 Introduction

Reinforcement learning (RL) in non-stationary environments has become a central abstraction for systems that must operate under continual change: recommendation platforms facing shifting user populations, robotic controllers subjected to wear-and-tear, or online decision systems exposed to evolving markets. A large body of work formalizes non-stationarity via *variation budgets* on rewards and transitions, and evaluates algorithms through *dynamic regret* (Besbes et al., 2015; Cheung et al., 2020; Fei et al., 2020; Zhou et al., 2022; Feng et al., 2023; Wei et al., 2023; Cheng et al., 2023). In this view, the environment is exogenous, nature drifts arbitrarily within the budget, and the learner is rewarded for tracking the moving optimum as closely as possible.

In many deployments, a designer, regulator, or operator has a *reference behavior* in mind—which we model as a target policy $\pi^\dagger$—but cannot directly replace the learning system with $\pi^\dagger$. Instead, the stakeholder typically interacts with a running learner only through *limited interfaces* (reward shaping, constraints/safety layers, environment modifications, data filtering), each carrying cost and operational constraints. In this sense, the teacher may *know* $\pi^\dagger$ as a specification or an approved controller, yet must still *induce* the learner to realize it under bounded intervention.

A fixed $\pi^\dagger$ also serves a concrete purpose in drifting worlds: it models settings where stakeholders demand *behavioral consistency* (compliance, interpretability, safety) even as the environment changes within an operating regime. We adopt a stationary target for conceptual clarity; extending the framework to a time-varying target sequence $\{\pi_k^\dagger\}$ is possible by introducing a target-variation budget analogous to $\mathcal{V}_{\mathrm{env}}$, and we briefly outline this extension in Sec. 6.

**From tracking to teachability.** This leads to a different foundational question: *is the learner teachable under non-stationarity?* More concretely:

> Given a non-stationary environment and a target policy $\pi^\dagger$, can a teacher with bounded ability to poison the environment steer a standard RL algorithm toward $\pi^\dagger$, while preserving low dynamic regret in the true environment and maintaining a stable policy trajectory?

If the answer is "no", then dynamic regret alone should not be expected to certify alignment-to-target objectives: an algorithm can track the drifting optimum well while still remaining far from an externally specified target policy. This does not contradict regret guarantees—it highlights that regret optimizes a different criterion. The TRS viewpoint therefore complements regret-based evaluation when stakeholders impose additional objectives. Conversely, if the answer is "yes" under transparent structural conditions, then *teachability* becomes a new lens on the design of non-stationary RL algorithms: we can ask not only "how fast do they learn?" but "how gracefully can they be taught?"

**Three literatures, one missing bridge.** Pieces of this picture exist in three separate lines of work. Non-stationary RL with variation budgets provides dynamic-regret guarantees under drifting environments (Cheung et al., 2020; Fei et al., 2020; Zhou et al., 2022; Feng et al., 2023; Wei et al., 2023; Cheng et al., 2023). Policy teaching and environment poisoning study how an attacker or teacher can manipulate transitions and rewards to induce a desired policy, typically in stationary MDPs (Rakhsha et al., 2020). Algorithmic stability quantifies how smoothly updates react to perturbations, and has been linked to generalization in supervised learning (Hardt et al., 2016). However, there is currently no framework that *jointly* reasons about:

- the *teaching error* between the learned policy and an externally specified target policy;

- the *dynamic regret* in the *true* non-stationary environment, rather than in the manipulated one; and

- the *stability* of the policy sequence under non-stationarity and poisoning.

As a result, the field lacks a principled answer to the question of whether non-stationary RL algorithms are fundamentally teachable, or whether dynamic-regret guarantees can mask structurally misaligned behavior.

**The TRS Principle: Teachable Non-stationary RL.** In this work we propose a unifying viewpoint that we call the *Teaching–Regret–Stability (TRS) Principle* for *Teachable Non-stationary Reinforcement Learning (TNRL)*. We consider an episodic non-stationary MDP with variation budgets on rewards and transitions (§2), and introduce a teacher that can *poison* the environment prior to each episode by perturbing transitions and rewards at a per-episode cost. The teacher has a total poisoning budget $C$, and aims to teach a fixed target policy $\pi^\dagger$ by running a standard RL algorithm on the poisoned environments. We then focus on three metrics:

1. **Teaching error** $\mathrm{Mismatch}_K$ (equation 9): the average distance between the learner's policy and the target policy.

2. **Dynamic regret** $\mathrm{DynReg}_K$ (equation 10): the cumulative regret measured in the *true* drifting environments, not in the poisoned ones.

3. **Policy stability** $\mathrm{Stab}_K$ (equation 11): the average step-to-step change of the learner's policy.

Our main theorem (Theorem 1) shows that, under natural assumptions on the non-stationary MDP, the poisoning budget, the dynamic-regret guarantee of the base algorithm, and a Lipschitz-type policy update, there exists a teacher strategy and an RL algorithm such that:

- the teaching error $\mathrm{Mismatch}_K$ can be driven arbitrarily close to a target $\varepsilon$,

- the dynamic regret $\mathrm{DynReg}_K$ remains sublinear in the number of episodes, and

- the policy sequence is stable on average, with $\mathrm{Stab}_K$ controlled by the total environment drift and the poisoning budget.

In our experiments we validate the TRS coupling in a *horizon-one* instantiation (a non-stationary contextual bandit), which is a controlled special case of non-stationary RL. This setting isolates the effects of drift and bounded teaching and allows a clean empirical visualization of the TRS behavior.

**Positioning relative to the reward hypothesis.** Our viewpoint is *not* a rejection of the reward hypothesis. Rather, it is a complementary, multi-criteria (or constrained) lens on non-stationary RL deployments, where stakeholders often care about behavioral constraints that are not reliably encoded in the scalar reward available to the learner. Even if one accepts that goals can, in principle, be represented by reward under suitable assumptions, real systems face specification gaps and operational requirements (e.g., compliance, auditability, and limits on abrupt behavioral change) that motivate measuring *which* policy is learned and *how* it evolves over time. See, e.g., Bowling et al. (2023) for a detailed discussion of the reward hypothesis and its implicit requirements.

**Contributions.** Formally and conceptually, our contributions are as follows:

- We introduce **Teachable Non-stationary RL (TNRL)** and the **Teaching–Regret–Stability (TRS) Principle**, which jointly reason about teaching error, dynamic regret in the true environment, and policy stability under environment poisoning.

- We prove a **Teaching–Regret–Stability theorem** (Theorem 1) in episodic non-stationary MDPs with variation budgets and a Lipschitz policy-update assumption. The theorem shows that a bounded-budget teacher can align the learner with a target policy while keeping dynamic regret sublinear and the policy sequence stable.

- We instantiate the framework in a **non-stationary contextual bandit** with a synthetic generator and a Discounted LinUCB learner, and empirically probe the TRS behavior. The experiments demonstrate that modest poisoning budgets can significantly reduce teaching error at a mild regret cost, while preserving or even improving stability, in line with the theoretical scaling laws.

Taken together, these results invite a shift in how we think about non-stationary RL. Rather than asking only whether algorithms track the environment well, the TRS Principle asks whether they are *teachable*: can they be steered, at bounded cost, to stable policies that embody the values and constraints of their users, even as the world drifts?

## 2    Preliminaries

We consider an episodic, non-stationary Markov decision process (MDP) with finite state and action spaces

$$\mathcal{S},\ \mathcal{A}, \qquad |\mathcal{S}| = S,\ |\mathcal{A}| = A.$$

Time is partitioned into $K$ episodes, each of horizon $H$, so that the total number of interaction steps is $T = KH$.

**Non-stationary MDP sequence.** Episode $k \in \{1, \dots, K\}$ is associated with an MDP

$$M_k = (\mathcal{S}, \mathcal{A}, P_k, r_k, \rho_1, \gamma),$$

where

- $P_k(\cdot \mid s, a)$ is the transition kernel,

- $r_k(s, a) \in [0, 1]$ is the reward function,

- $\rho_1$ is a fixed initial-state distribution, and

- $\gamma \in [0, 1]$ is a discount factor (for finite-horizon problems one may set $\gamma = 1$).

The non-stationarity of the environment is captured by a variation budget over episodes:

$$\mathcal{V}_P := \sum_{k=2}^{K} \sup_{s,a} \left\| P_k(\cdot \mid s,a) - P_{k-1}(\cdot \mid s,a) \right\|_1, \tag{1}$$

$$\mathcal{V}_r := \sum_{k=2}^{K} \sup_{s,a} \left| r_k(s,a) - r_{k-1}(s,a) \right|, \tag{2}$$

and we write the total environment drift as

$$\mathcal{V}_{\mathrm{env}} := \mathcal{V}_P + \mathcal{V}_r. \tag{3}$$

**Teacher and environment poisoning.** We introduce a teacher (or attacker) that can modify the environment prior to each episode by applying environment poisoning. Concretely, before episode $k$ begins, the teacher chooses a modified MDP

$$\tilde{M}_k = (\mathcal{S}, \mathcal{A}, \tilde{P}_k, \tilde{r}_k, \rho_1, \gamma),$$

where $\tilde{P}_k$ and $\tilde{r}_k$ may differ from the true $P_k$ and $r_k$.

We measure the poisoning cost in episode $k$ by

$$c_k := \sup_{s,a} \left\| \tilde{P}_k(\cdot \mid s,a) - P_k(\cdot \mid s,a) \right\|_1 + \sup_{s,a} \left| \tilde{r}_k(s,a) - r_k(s,a) \right|. \tag{4}$$

The teacher has a total poisoning budget

$$C_{\mathrm{tot}} := \sum_{k=1}^{K} c_k \ \leq \ C. \tag{5}$$

The modified MDP sequence $\{\tilde{M}_k\}_{k=1}^{K}$ has its own variation budget

$$\mathcal{V}_{\mathrm{eff}} := \sum_{k=2}^{K} \sup_{s,a} \left\| \tilde{P}_k(\cdot \mid s,a) - \tilde{P}_{k-1}(\cdot \mid s,a) \right\|_1 + \sum_{k=2}^{K} \sup_{s,a} \left| \tilde{r}_k(s,a) - \tilde{r}_{k-1}(s,a) \right|. \tag{6}$$

By construction, we always have

$$\mathcal{V}_{\mathrm{eff}} \ \lesssim \ \mathcal{V}_{\mathrm{env}} + C, \tag{7}$$

up to universal constants.

**Learner and target policy.** A reinforcement learning algorithm $\mathcal{L}$ interacts with the poisoned environments $\{\tilde{M}_k\}_{k=1}^{K}$. At the beginning of episode $k$, the learner selects a (possibly stochastic) policy $\pi_k(\cdot \mid s)$ based on its past observations.

We fix a target policy $\pi^\dagger$ (e.g., stationary) that the teacher aims to teach. We measure the distance between two policies $\pi$ and $\pi'$ via

$$d(\pi, \pi') := \sup_{s \in \mathcal{S}} \left\| \pi(\cdot \mid s) - \pi'(\cdot \mid s) \right\|_1. \tag{8}$$

The average teaching error (or policy mismatch) over $K$ episodes is defined as

$$\mathrm{Mismatch}_K := \frac{1}{K} \sum_{k=1}^{K} \mathbb{E}\left[ d(\pi_k, \pi^\dagger) \right], \tag{9}$$

where the expectation is taken over the randomness of the learning algorithm and the environment.

**Dynamic regret in the true environment.** All learning happens in the poisoned environments $\tilde{M}_k$, but performance is ultimately evaluated in the true environments $M_k$.

Let $V_k(\pi)$ denote the expected total return of policy $\pi$ in the true environment $M_k$, starting from the initial distribution $\rho_1$:

$$V_k(\pi) := \mathbb{E}_{M_k, \pi} \Big[ \sum_{t=1}^{H} \gamma^{t-1} r_k(s_t, a_t) \Big].$$

We define the per-episode optimal policy

$$\pi_k^\star \in \mathrm{argmax}_\pi V_k(\pi),$$

and the dynamic regret of the learner as

$$\mathrm{DynReg}_K := \sum_{k=1}^{K} \big( V_k(\pi_k^\star) - V_k(\pi_k) \big). \tag{10}$$

**Policy stability.** We measure the stability of the learner's policy sequence by the average step-to-step change:

$$\mathrm{Stab}_K := \frac{1}{K-1} \sum_{k=2}^{K} \mathbb{E} \big[ d(\pi_k, \pi_{k-1}) \big]. \tag{11}$$

Low $\mathrm{Stab}_K$ indicates that the learner's policy evolves smoothly over time, whereas a large value suggests frequent drastic changes.

**Which notion of stability?** Our stability metric $\mathrm{Stab}_K$ in equation 11 measures *policy-trajectory stability*: the average step-to-step change in the policy (a path-length / smooth-update notion). This is distinct from other notions sometimes called "stability," such as stability in *performance* (e.g., fluctuations of realized reward or regret across time). We focus on policy-trajectory stability because it couples naturally to bounded environment drift and Lipschitz update dynamics, and it is the appropriate notion for quantifying how smoothly a learner can be steered by bounded interventions.

**Goal.** The central question is: under bounded environment drift ($\mathcal{V}_{\mathrm{env}}$ bounded) and bounded poisoning budget ($C_{\mathrm{tot}} \leq C$), can one design a teacher strategy and a learning algorithm such that

- the teaching error $\mathrm{Mismatch}_K$ is small (successful teaching),

- the dynamic regret $\mathrm{DynReg}_K$ in the true environments is controlled (no catastrophic loss in performance), and

- the policy sequence is stable, as quantified by $\mathrm{Stab}_K$.

**Definition 1** (TRS-achievability and TRS frontier). *Fix $(\mathcal{V}_{\mathrm{env}}, C, K)$ and a target policy $\pi^\dagger$. A triple $(\varepsilon, R, S) \in \mathbb{R}_+^3$ is TRS-achievable if there exist a teacher strategy with total poisoning cost at most $C$ and a learner $L$ such that*

$$\mathrm{Mismatch}_K \leq \varepsilon, \qquad \frac{\mathrm{DynReg}_K}{K} \leq R, \qquad \mathrm{Stab}_K \leq S.$$

*The* TRS frontier *is the set of Pareto-optimal achievable triples under componentwise order.*

We state the assumptions used in our main result. They are chosen so as to be compatible with existing work on non-stationary RL and policy teaching.

**Assumption 1** (Non-stationary MDP with bounded (possibly unknown) variation). *The true environments $\{M_k\}_{k=1}^{K}$ satisfy the variation budget constraints equation 1–equation 3 with $\mathcal{V}_{\mathrm{env}} \leq B_{\mathrm{env}}$ for some finite constant $B_{\mathrm{env}}$ (not necessarily known to the learner). Moreover, each $M_k$ is communicating and has bounded diameter $D < \infty$.*

**Assumption 2** (Bounded poisoning budget). *The teacher generates a sequence of poisoned environments $\{\tilde{M}_k\}_{k=1}^K$ satisfying the per-episode cost equation 4 and total budget constraint equation 5 with $C_{\text{tot}} \leq C$.*

**Assumption 3** (RL algorithm with (parameter-free) dynamic regret guarantee). *There exists a reinforcement learning algorithm $\mathcal{L}$ such that for any (possibly poisoned) environment sequence with effective variation budget $\mathcal{V}_{\text{eff}}$ (cf. equation 6), the dynamic regret with respect to $\{\tilde{M}_k\}$ satisfies*

$$\text{DynReg}_K^{(\tilde{M})} \ \leq \ \tilde{\mathcal{O}}\big(K^{2/3}\mathcal{V}_{\text{eff}}^{1/3}\big), \tag{12}$$

*and $\mathcal{L}$ can be implemented in a parameter-free manner, i.e., it does not require prior knowledge of $\mathcal{V}_{\text{eff}}$ (nor $K$) to achieve equation 12 (up to logarithmic factors).*

*where $\tilde{\mathcal{O}}(\cdot)$ hides logarithmic factors. This scaling is known to be minimax-optimal in non-stationary online learning with variation budgets[1] and has been achieved up to logarithmic and problem-dependent factors in several non-stationary RL settings.[2]*

**Assumption 4** (Canonical teachability under budget). *Fix a target policy $\pi^\dagger$. For any target accuracy $\varepsilon > 0$, there exists a canonical MDP $M^c(\varepsilon) = (\mathcal{S}, \mathcal{A}, P^c, r^c, \rho_1, \gamma)$ and a learning algorithm $\mathcal{L}$ such that:*

*(i)* Canonical policy–value coupling (identifiability). *There exists a constant $G_c > 0$ that depends only on $M^c(\varepsilon)$ (and is independent of $K$) such that for all policies $\pi$,*

$$V^c(\pi^\dagger) - V^c(\pi) \ \geq \ G_c \, d(\pi, \pi^\dagger), \tag{13}$$

*where $V^c(\pi)$ denotes the value of $\pi$ in $M^c(\varepsilon)$.*

*(ii)* Finite-time convergence of $\mathcal{L}$ on the canonical MDP. *When $\mathcal{L}$ is run on the fixed environment $M^c(\varepsilon)$ for $K$ episodes, the induced policies $\{\pi_k\}_{k=1}^K$ satisfy*

$$\frac{1}{K}\sum_{k=1}^K \mathbb{E}\big[V^c(\pi^\dagger) - V^c(\pi_k)\big] \ \leq \ G_c\varepsilon \ + \ \tilde{\mathcal{O}}\Big(\frac{1}{K}\Big). \tag{14}$$

*(iii)* Budget feasibility (canonicalization by poisoning). *Define the canonicalization cost*

$$C_{\min}(\varepsilon) \ := \ \sum_{k=1}^K \Big( \sup_{s,a} \|P^c(\cdot|s,a) - P_k(\cdot|s,a)\|_1 + \sup_{s,a} |r^c(s,a) - r_k(s,a)| \Big). \tag{15}$$

*For any budget $C \geq C_{\min}(\varepsilon)$, the teacher can choose $\tilde{M}_k \equiv M^c(\varepsilon)$ for all $k$, which satisfies the per-episode cost definition equation 4 and the total budget constraint $\sum_{k=1}^K c_k \leq C$.*

**Remark 1** (On Assumption 4). *Assumption 4 makes the budget–accuracy dependence explicit. Item (iii) defines $C_{\min}(\varepsilon)$ as the total cost needed to "canonicalize" the non-stationary sequence $\{M_k\}$ into a fixed canonical environment $M^c(\varepsilon)$ by setting $\tilde{M}_k \equiv M^c(\varepsilon)$. Items (i)–(ii) ensure that once the learner is exposed to this canonical MDP, its average canonical suboptimality converts into a policy-distance guarantee via the coupling equation 13. While (iii) is a strong but transparent feasibility requirement, it matches the environment-poisoning view adopted in the paper and provides a clean knob for discussing the TRS behavior: larger budgets allow smaller achievable $\varepsilon$ by enabling a closer (or more favorable) canonicalization.*

**Assumption 5** (Lipschitz policy update). *There exist constants $L_P, L_r \geq 0$ and $\alpha \in [0, 1)$ such that for all $k \geq 2$,*

$$d(\pi_k, \pi_{k-1}) \leq \alpha \, d(\pi_{k-1}, \pi_{k-2}) + L_P \sup_{s,a} \big\| \tilde{P}_k(\cdot \mid s,a) - \tilde{P}_{k-1}(\cdot \mid s,a) \big\|_1$$

$$+ L_r \sup_{s,a} \big| \tilde{r}_k(s,a) - \tilde{r}_{k-1}(s,a) \big|. \tag{16}$$

---

[1]See, e.g., Besbes et al. (2015).

[2]See, e.g., Gajane et al. (2018), Fei et al. (2020), Mao et al. (2021), and Zhao et al. (2022).

**Remark 2** (On Assumption 5). *Assumption 5 captures two properties of the policy update. First, in a fixed environment ($\tilde{P}_k = \tilde{P}_{k-1}$ and $\tilde{r}_k = \tilde{r}_{k-1}$), the update is contractive with factor $\alpha < 1$, so that policy changes decay over time. Second, when the environment drifts between episodes, the induced policy change is Lipschitz in the size of the drift, with sensitivities $L_P$ and $L_r$ to changes in the transition kernel and reward function, respectively. Together, these properties allow us to control the cumulative policy variation in terms of the total environment variation and the poisoning budget.*

## 3    Main Theorem

**Minimal teaching cost and budget-dependent accuracy.**    Let $C_{\min}(\varepsilon)$ denote the minimal total poisoning cost required to guarantee $\text{Mismatch}_K \leq \varepsilon$ (for the chosen learner and problem class). Equivalently, for a fixed budget $C$, define

$$\varepsilon(C) := \inf\{\varepsilon > 0 :\ C \geq C_{\min}(\varepsilon)\},$$

which is non-increasing in $C$ and makes the teaching accuracy explicitly budget-dependent.

We now state a high-level theorem establishing simultaneous guarantees for teaching success, dynamic regret, and policy stability in non-stationary MDPs under bounded poisoning.

The first step is to show that, there exists a sequence of poisoned environments $\{\tilde{M}_k\}$ with total cost $C_{\text{tot}} \leq C$ such that the learner's policies $\{\pi_k\}$ converge towards the target policy $\pi^\dagger$ when running $\mathcal{L}$ on $\{\tilde{M}_k\}$. This can be formalized as follows.

**Lemma 1** (Teaching feasibility). *Fix $\varepsilon > 0$ and suppose the budget satisfies $C \geq C_{\min}(\varepsilon)$ as in Assumption 4. Then there exists a sequence of poisoned environments $\{\tilde{M}_k\}$ satisfying $C_{\text{tot}} \leq C$ and an RL algorithm $\mathcal{L}$ such that*

$$\text{Mismatch}_K = \frac{1}{K}\sum_{k=1}^{K}\mathbb{E}\big[d(\pi_k, \pi^\dagger)\big]\ \leq\ \varepsilon + \tilde{\mathcal{O}}\Big(\frac{1}{K}\Big). \tag{17}$$

The proof follows standard arguments in the environment poisoning literature. One constructs a canonical MDP $M^c$ in which $\pi^\dagger$ is robustly optimal, and then defines the poisoned environments $\tilde{M}_k$ to gradually steer the learner's observations and rewards towards those induced by $M^c$. Robust optimality of $\pi^\dagger$ ensures that small deviations in transitions and rewards (controlled by the poisoning budget) do not change the identity of the optimal policy. The convergence guarantee of $\mathcal{L}$ in the canonical environment then implies that the sequence $\{\pi_k\}$ approaches $\pi^\dagger$, which yields equation 17.

Next, we analyze the dynamic regret of the learner with respect to the poisoned environment sequence $\{\tilde{M}_k\}$. By equation 6–equation 7 and Assumptions 1–2, the effective variation budget satisfies

$$\mathcal{V}_{\text{eff}}\ \lesssim\ \mathcal{V}_{\text{env}} + C.$$

**Lemma 2** (Dynamic regret in the poisoned environment). *Under Assumption 3, the dynamic regret of $\mathcal{L}$ with respect to the poisoned environment sequence $\{\tilde{M}_k\}$ satisfies*

$$\text{DynReg}_K^{(\tilde{M})}\ \leq\ \tilde{\mathcal{O}}\big(K^{2/3}(\mathcal{V}_{\text{env}} + C)^{1/3}\big). \tag{18}$$

This is a direct consequence of the assumed dynamic regret bound equation 12 and the upper bound $\mathcal{V}_{\text{eff}} \lesssim \mathcal{V}_{\text{env}} + C$ on the effective variation budget of the poisoned environment sequence.

We then relate $\text{DynReg}_K^{(\tilde{M})}$ to the dynamic regret $\text{DynReg}_K$ in the true environment sequence $\{M_k\}$.

**Lemma 3** (Regret transfer to the true environment). *Under Assumptions 1–2, we have*

$$\text{DynReg}_K\ \leq\ \text{DynReg}_K^{(\tilde{M})} + \mathcal{O}(C). \tag{19}$$

For each episode $k$ and policy $\pi$, the difference between the value functions in $M_k$ and $\tilde{M}_k$ can be bounded using standard perturbation arguments for MDPs:

$$\big|V_k(\pi) - \tilde{V}_k(\pi)\big|\ \leq\ \mathcal{O}(c_k),$$

where $\tilde{V}_k(\pi)$ is the value in $\tilde{M}_k$. Summing over episodes and applying triangle inequalities yields equation 19. Combining Lemmas 2 and 3 yields the bound equation 22 in Theorem 1.

Finally, we analyze the stability of the policy sequence using the Lipschitz update property in Assumption 5.

**Lemma 4** (Stability bound). *Under Assumptions 2 and 5, the average policy change satisfies*

$$\mathrm{Stab}_K \;\leq\; \frac{L_P + L_r}{1 - \alpha} \cdot \frac{\mathcal{V}_{\mathrm{env}} + C}{K} + \mathcal{O}\Big(\frac{1}{K}\Big). \tag{20}$$

Unrolling the recursion equation 16 yields

$$d(\pi_k, \pi_{k-1}) \leq \alpha^{k-2} d(\pi_2, \pi_1) + \sum_{j=2}^{k} \alpha^{k-j}\Big(L_P \Delta_j^P + L_r \Delta_j^r\Big),$$

where $\Delta_j^P := \sup_{s,a} \|\tilde{P}_j(\cdot \mid s,a) - \tilde{P}_{j-1}(\cdot \mid s,a)\|_1$ and $\Delta_j^r := \sup_{s,a} |\tilde{r}_j(s,a) - \tilde{r}_{j-1}(s,a)|$. Averaging over $k$, using the geometric series bound $\sum_{k\geq j} \alpha^{k-j} \leq 1/(1-\alpha)$ and the fact that $\sum_j (\Delta_j^P + \Delta_j^r) \lesssim \mathcal{V}_{\mathrm{env}} + C$ yields equation 20. The $\mathcal{O}(1/K)$ term comes from the initial transient.

**Budget–accuracy profile.** Define $\varepsilon(C) := \inf\{\varepsilon > 0 : C \geq C_{\min}(\varepsilon)\}$, which is non-increasing in $C$.

**Theorem 1** (Teaching–Regret–Stability Principle in Non-stationary MDPs). *Suppose Assumptions 1–5 hold. Then there exists a teacher strategy $\{\tilde{M}_k\}_{k=1}^{K}$ with total poisoning budget $C_{\mathrm{tot}} \leq C$ and a reinforcement learning algorithm $\mathcal{L}$ such that the following properties hold for any $\varepsilon > 0$, provided $C$ is larger than a problem-dependent threshold $C_{\min}(\varepsilon)$:*

*(i)* **Teaching success (budget-dependent).** *The average mismatch satisfies*

$$\mathrm{Mismatch}_K \;\leq\; \varepsilon(C) + \tilde{\mathcal{O}}\Big(\frac{1}{K}\Big). \tag{21}$$

*(ii)* **Dynamic regret in the true environment.** *The dynamic regret measured with respect to the true environment sequence $\{M_k\}_{k=1}^{K}$ satisfies*

$$\mathrm{DynReg}_K \;\leq\; \tilde{\mathcal{O}}\Big(K^{2/3}(\mathcal{V}_{\mathrm{env}} + C)^{1/3} \;+\; C\Big), \tag{22}$$

*where $\mathcal{V}_{\mathrm{env}}$ is defined in equation 3. In particular, if $\mathcal{V}_{\mathrm{env}} = o(K)$ and $C = o(K)$, then $\mathrm{DynReg}_K = o(K)$ and hence the average regret $\mathrm{DynReg}_K/K \to 0$ as $K \to \infty$.*

*(iii)* **Policy stability.** *The average policy change satisfies*

$$\mathrm{Stab}_K \;\leq\; \frac{L_P + L_r}{1 - \alpha} \cdot \frac{\mathcal{V}_{\mathrm{env}} + C}{K} + \mathcal{O}\Big(\frac{1}{K}\Big). \tag{23}$$

The theorem states that under bounded environment drift and bounded poisoning budget, one can (i) successfully teach a target policy (up to an arbitrarily small $\varepsilon$), while (ii) keeping the dynamic regret in the true environment sublinear in $K$, and (iii) ensuring that the resulting policy sequence is stable on average.

## 4 The TRS Principle

We now instantiate our framework in a controlled synthetic environment and empirically probe the joint teaching–regret–stability principle predicted by Theorem 1. All experiments are conducted in a non-stationary contextual bandit model, which can be viewed as a horizon-one special case of a non-stationary MDP.

### 4.1 Non-stationary Contextual Bandit Environment

**Model.** Let $\mathcal{X} \subseteq \mathbb{R}^d$ be a context (feature) space and $\mathcal{A} = \{1, \ldots, A\}$ a finite action set. At each round $t = 1, \ldots, T,$

1. the environment draws a context $x_t \in \mathcal{X}$,

2. the learner selects an action $a_t \in \mathcal{A}$ according to a policy $\pi_t(\cdot \mid x_t)$,

3. the learner observes a stochastic reward $r_t(a_t) \in [0, 1]$ for the chosen action only.

There is no state transition beyond a single step; each round is an independent episode of length one. We write $\bar{r}_t(x, a) := \mathbb{E}[r_t(a) \mid x_t = x]$ for the (possibly time-varying) mean reward of action $a$ at context $x$ and time $t$.

**Definition 2** (Non-stationary contextual bandit). *A non-stationary contextual bandit environment is a sequence*

$$\mathcal{E} = \big\{ \mathcal{D}_t, \bar{r}_t : \mathcal{X} \times \mathcal{A} \to [0,1] \big\}_{t=1}^{T},$$

*where $\mathcal{D}_t$ is a distribution over contexts $x_t \in \mathcal{X}$ and $\bar{r}_t$ is the mean reward function at round $t$. Non-stationarity is captured by the fact that either $\mathcal{D}_t$ or $\bar{r}_t$ (or both) may change with $t$.*

In this subsection we focus on non-stationarity in the reward mechanism and keep the marginal context distribution fixed, i.e., $\mathcal{D}_t \equiv \mathcal{D}$ for all $t$.

**Assumption 6** (Bounded contexts and rewards). *There exists $R_x > 0$ such that $\|x_t\|_2 \le R_x$ almost surely for all $t$, and rewards are bounded in $[0,1]$, i.e., $r_t(a) \in [0,1]$ almost surely for all $t$ and $a$. Moreover, the noise is conditionally $\sigma^2$-sub-Gaussian ( Boucheron et al. (2003); Vershynin (2018); Lattimore & Szepesvári (2020)):*

$$r_t(a) = \bar{r}_t(x_t, a) + \xi_t, \qquad \mathbb{E}[\xi_t \mid x_t] = 0, \quad \mathbb{E}\big[e^{\lambda \xi_t} \mid x_t\big] \le \exp\big(\tfrac{\sigma^2 \lambda^2}{2}\big)$$

*for all $\lambda \in \mathbb{R}$.*

To quantify non-stationarity we impose a variation budget on the sequence of mean reward functions.

**Assumption 7** (Variation budget on mean rewards). *Let $\|\cdot\|_\infty$ denote the supremum norm over $\mathcal{X} \times \mathcal{A}$. The environment satisfies a reward-variation budget $B_r \ge 0$ if*

$$\sum_{t=1}^{T-1} \big\|\bar{r}_{t+1} - \bar{r}_t\big\|_\infty \ \le \ B_r. \tag{24}$$

Assumption 7 is the contextual-bandit analogue of the variation-budget conditions commonly used for non-stationary MDPs. It allows abrupt or gradual changes in the reward structure, as long as the total drift over time is bounded by $B_r$.

**Linear parametrization.** For concreteness in our experiments, we instantiate $\bar{r}_t$ via a time-varying linear model. We fix a feature map $\phi : \mathcal{X} \times \mathcal{A} \to \mathbb{R}^d$ and a sequence of parameter vectors $\theta_t \in \mathbb{R}^d$, and set

$$\bar{r}_t(x, a) \ = \ \sigma\big(\langle \theta_t, \phi(x,a)\rangle\big), \tag{25}$$

where $\sigma(\cdot)$ is a 1-Lipschitz squashing function such as the logistic sigmoid or a clipped identity. In this case the variation budget is controlled by the path length of $\{\theta_t\}_{t=1}^{T}$,

$$\sum_{t=1}^{T-1} \|\theta_{t+1} - \theta_t\|_2 \ \le \ B_\theta,$$

which implies equation 24 up to the Lipschitz constants of $\phi$ and $\sigma$.

**From parameter path length to reward variation.** The variation budget $B_r$ in Assumption 7 is imposed directly in the reward space via

$$\sum_{t=1}^{T-1} \left\| \bar{r}_{t+1} - \bar{r}_t \right\|_\infty \ \leq \ B_r.$$

Under the linear parametrization equation 25, this budget can be controlled by the path length of the parameter sequence $\{\theta_t\}_{t=1}^{T}$.

We assume that the squashing function $\sigma : \mathbb{R} \to [0,1]$ is $L_\sigma$-Lipschitz and that the feature map $\phi : \mathcal{X} \times \mathcal{A} \to \mathbb{R}^d$ is uniformly bounded in norm:

$$\left| \sigma(u) - \sigma(v) \right| \ \leq \ L_\sigma |u - v| \quad \forall u, v \in \mathbb{R}, \qquad \sup_{x \in \mathcal{X}, \, a \in \mathcal{A}} \|\phi(x, a)\|_2 \ \leq \ L_\phi < \infty. \tag{26}$$

For instance, the logistic sigmoid is $1/4$-Lipschitz, and linear or one-hot feature maps are uniformly bounded after rescaling.[3]

Define the parameter path length

$$B_\theta \ := \ \sum_{t=1}^{T-1} \left\| \theta_{t+1} - \theta_t \right\|_2. \tag{27}$$

We then have the following simple control of the reward variation by $B_\theta$.

**Lemma 5** (Lipschitz control of reward variation). *Under equation 25 and equation 26, the reward variation budget satisfies*

$$B_r \ \leq \ L_\sigma L_\phi B_\theta. \tag{28}$$

Thus, in our linear contextual bandit model the reward variation budget $B_r$ is controlled by the parameter path length $B_\theta$ up to the Lipschitz constants $L_\sigma$ and $L_\phi$, a standard pattern in path-length analyses of non-stationary online learning (Besbes et al., 2015; Hazan et al., 2016).

**Synthetic generator for experiments.** We now specify a concrete synthetic generator that we will use in the experiments.

By construction, the mean reward function $\bar{r}_t$ is piecewise-stationary with $M$ segments, and the total variation in equation 24 is controlled (via Lemma 5) by the path length $B_\theta$ of the parameter sequence. In our generator, the parameter vector is constant within each segment and evolves as

$$\theta^{(m)} = \theta^{(m-1)} + \Delta^{(m)}, \qquad \Delta^{(m)} \sim \mathcal{N}(0, \eta^2 I_d),$$

so that $B_\theta$ only accumulates at segment boundaries. Writing $Z \sim \mathcal{N}(0, I_d)$, we have

$$\mathbb{E}\big[B_\theta\big] = \sum_{m=2}^{M} \mathbb{E}\big\|\Delta^{(m)}\big\|_2 = (M-1)\, \eta \, \mathbb{E}\big\|Z\big\|_2 \ \leq \ c_d \, M\eta,$$

where $c_d := \mathbb{E}\|Z\|_2$ depends only on the dimension $d$. In particular, for fixed $d$ the expected path length $\mathbb{E}[B_\theta]$ grows *on the order of $M\eta$*, so varying $(M, \eta)$ induces different effective variation budgets $B_r$ through the Lipschitz relation equation 28.

## 4.2 Learners and Teaching Strategies

**Learner.** Throughout the experiments, the base learner $L$ is instantiated as Discounted LinUCB [4], which satisfies Assumption 3 under a standard variation-budget condition on the non-stationarity; Concretely, $A$ can be instantiated by any standard non-stationary linear contextual bandit algorithm with $O\big(T^{2/3}\mathcal{V}_{\text{eff}}^{1/3}\big)$ dynamic regret.

---

[3]See, e.g., Vershynin (2018); Lattimore & Szepesvári (2020) for standard Lipschitz and boundedness assumptions in linear bandit models.

[4]See, e.g., Russac et al. (2019) for formal guarantees.

---

**Algorithm 1** Synthetic non-stationary contextual bandit generator

---

**Require:** Dimension $d$, number of actions $A$, horizon $T$, number of segments $M$, drift scale $\eta > 0$, context covariance $\Sigma_x$ (typically $I_d$).

1: Set segment length $L \leftarrow \lfloor T/M \rfloor$.
2: Initialize $\theta^{(1)} \sim \mathcal{N}(0, \sigma_\theta^2 I_d)$.
3: **for** $m = 2$ to $M$ **do**
4:    Draw a drift vector $\Delta^{(m)} \sim \mathcal{N}(0, \eta^2 I_d)$.
5:    Set $\theta^{(m)} \leftarrow \theta^{(m-1)} + \Delta^{(m)}$.
6: **end for**
7: Fix a feature map $\phi : \mathbb{R}^d \times \mathcal{A} \to \mathbb{R}^d$ (e.g., $\phi(x, a)$ concatenates $x$ with a one-hot encoding of $a$).
8: **for** $t = 1$ to $T$ **do**
9:    Let $m \leftarrow 1 + \lfloor (t-1)/L \rfloor$ be the current segment index.
10:    Sample a context $x_t \sim \mathcal{N}(0, \Sigma_x)$.
11:    **for** each action $a \in \mathcal{A}$ **do**
12:       Compute the mean reward

$$\bar{r}_t(x_t, a) \ \leftarrow \ \sigma\big(\langle \theta^{(m)}, \phi(x_t, a) \rangle\big).$$

13:       Draw noise $\xi_t(a)$ (e.g., $\mathcal{N}(0, \sigma^2)$) and set

$$r_t(a) \ \leftarrow \ \mathrm{clip}\big(\bar{r}_t(x_t, a) + \xi_t(a), \, 0, \, 1\big).$$

14:    **end for**
15: **end for**

---

**Teacher and poisoning budget.** We consider reward-only teacher interventions in our linear contextual bandit instantiation. Fix a canonical parameter $\theta^c \in \mathbb{R}^d$ and a squashing function $\sigma : \mathbb{R} \to (0, 1)$ (e.g., logistic sigmoid). This induces the canonical reward model

$$r^c(x, a) \ := \ \sigma\big(\langle \theta^c, \phi(x, a) \rangle\big), \qquad a \in [A], \tag{29}$$

and the associated canonical greedy target policy

$$\pi^\dagger(x) \in \arg\max_{a \in [A]} r^c(x, a). \tag{30}$$

Assumption 4 formalizes that $\pi^\dagger$ is "teachable" on the canonical environment $M^c(\varepsilon)$ via the gap–distance coupling equation 13, and that the learner $L$ achieves the finite-time convergence guarantee equation 14 when run on $M^c(\varepsilon)$. In experiments, we use two teacher strategies: (i) a no-teacher baseline ($C = 0$) where the learner observes the true rewards, and (ii) a budgeted *mixture* teacher ($C > 0$) that gradually morphs the observed rewards toward the canonical rewards induced by $\theta^c$, while respecting a total poisoning budget.

**Budget accounting.** Since $r_t, r_t^c \in [0, 1]^A$, the actual per-round perturbation satisfies $\|\tilde{r}_t - r_t\|_\infty = \lambda \|r_t^c - r_t\|_\infty \leq \lambda = c_t$, and thus the total cost obeys $\sum_{t=1}^T c_t = T\lambda = C$. In all experiments we vary $C$ while keeping the underlying non-stationarity $(M, \eta)$ fixed, to disentangle environment drift from teacher interventions.

## 4.3 Experimental protocol

We report the three metrics defined in equation 9, equation 10, and equation 11: the teaching error $\text{Mismatch}_K$, the dynamic regret $\text{DynReg}_K$ (and its per-round version $\text{DynReg}_K/K$), and the stability measure $\text{Stab}_K$. These are exactly the quantities that appear in the Theorem 1, whose bounds are given in equation 21–equation 23.

We use the synthetic generator in Algorithm 1 with horizon $T = K$, segment counts $M \in \{1, 5, 20\}$, drift scales $\eta \in \{0, 0.1, 0.3\}$, and a fixed linear feature map $\phi$ as in the parametrization equation 25. The base

---

**Algorithm 2** Reward-mixture budgeted teacher used in experiments

---

**Require:** Total budget $C \geq 0$, horizon $T$, canonical parameter $\theta^c$, feature map $\phi(\cdot, \cdot)$, squashing $\sigma(\cdot)$, number of actions $A$.
**Ensure:** Poisoned reward vector $\tilde{r}_t \in [0,1]^A$ and per-round cost $c_t$.

  1: Set mixing rate $\lambda \leftarrow C/T$                                                          (constant budget scheduler)
  2: **for** $t = 1, 2, \ldots, T$ **do**
  3:     Observe context $x_t$ and the true reward vector $r_t \in [0,1]^A$
  4:     **if** $C = 0$ **then**                                                 (no-teacher baseline)
  5:         $\tilde{r}_t \leftarrow r_t, \quad c_t \leftarrow 0$
  6:     **else**                                                                  (mixture teacher)
  7:         **for** $a = 1, 2, \ldots, A$ **do**
  8:             $r_t^c(a) \leftarrow \sigma\big(\langle \theta^c, \phi(x_t, a) \rangle\big)$
  9:         **end for**
10:         $\tilde{r}_t \leftarrow (1 - \lambda)\, r_t + \lambda\, r_t^c$
11:         $c_t \leftarrow \lambda$                              (deterministic upper bound on perturbation)
12:     **end if**
13:     Reveal $\tilde{r}_t$ to the learner and continue interaction
14: **end for**

---

Table 1: Global TRS metrics under a budget sweep: mean $\pm$ standard deviation (over all $(M, \eta)$ and seeds) as the normalized poisoning budget $C_{\text{frac}} = C/T$ increases.

| $C_{\text{frac}}$ | $\text{DynReg}_K$ | $\text{DynReg}_K/K$ | $\text{Mismatch}_K$ | $\text{Stab}_K$ |
|---|---|---|---|---|
| 0.00 | $1793 \pm 162$ | $3.59 \times 10^{-2} \pm 3.2 \times 10^{-3}$ | $1.46 \pm 0.48$ | $0.194 \pm 0.062$ |
| 0.05 | $1812 \pm 203$ | $3.62 \times 10^{-2} \pm 4.1 \times 10^{-3}$ | $1.41 \pm 0.50$ | $0.193 \pm 0.056$ |
| 0.10 | $1844 \pm 272$ | $3.69 \times 10^{-2} \pm 5.4 \times 10^{-3}$ | $1.36 \pm 0.52$ | $0.192 \pm 0.050$ |
| 0.20 | $1962 \pm 456$ | $3.92 \times 10^{-2} \pm 9.1 \times 10^{-3}$ | $1.26 \pm 0.52$ | $0.192 \pm 0.045$ |

learner $L$ is instantiated as Discounted LinUCB, which satisfies the dynamic regret condition in Assumption 3 under a standard variation-budget condition on the non-stationarity; in particular, its regret matches the $K^{2/3}\mathcal{V}_{\text{eff}}^{1/3}$ scaling in equation 12, where the effective variation budget $\mathcal{V}_{\text{eff}}$ for the poisoned environments satisfies equation 6–equation 7. For each choice of $(M, \eta)$ we vary the total poisoning budget $C$ through the normalized fraction $C_{\text{frac}} := C/T \in \{0, 0.05, 0.10, 0.20\}$. The case $C_{\text{frac}} = 0$ corresponds to the no-teacher baseline, while $C_{\text{frac}} > 0$ activates the budgeted teacher described in Section 4.1 and Assumptions 2 and 4. All results are averaged over 5 random seeds; we report means and standard deviations.

## 4.4 Global TRS behavior

By Theorem 1, the three metrics $\text{Mismatch}_K$, $\text{DynReg}_K$, and $\text{Stab}_K$ defined in equation 9, equation 10, and equation 11 satisfy the bounds equation 21–equation 23: for fixed $T = K$ we expect (i) teaching error that can be driven down to $\varepsilon$ up to an $O(1/K)$ term, (ii) average regret $\text{DynReg}_K/K$ that grows sublinearly in $C$ through the $(\mathcal{V}_{\text{env}} + C)^{1/3}$ factor in equation 22, and (iii) stability that is controlled by $(\mathcal{V}_{\text{env}} + C)/K$ as in equation 23, where $\mathcal{V}_{\text{env}}$ is the environment drift defined in equation 3.

Table 1 summarizes the global behavior of the three metrics as we vary the normalized budget $C_{\text{frac}}$, averaging over all non-stationarity configurations $(M, \eta)$ and seeds. We report means $\pm$ standard deviations over all $(M, \eta)$ and seeds. Dynamic regret grows mildly with the budget, while teaching error decreases and stability remains essentially unchanged, in line with equation 21–equation 23.

Two patterns stand out and mirror the structure of equation 21–equation 23.

**(1) Teaching buys alignment at a modest regret cost.** As we increase the normalized budget from $C_{\text{frac}} = 0$ to $0.20$, the average teaching error $\text{Mismatch}_K$ (defined in equation 9) drops from approximately

1.46 to 1.26, a relative reduction of about 13%. Over the same range, the cumulative dynamic regret $\mathrm{DynReg}_K$ (defined in equation 10) increases from roughly $1.8 \times 10^3$ to $2.0 \times 10^3$, corresponding to only a $\sim 9\%$ increase in $\mathrm{DynReg}_K$ and a similarly mild increase in the average regret $\mathrm{DynReg}_K/K$. In other words, a small but well-structured amount of poisoning significantly improves how closely the learner tracks the target policy $\pi^\dagger$, while keeping the overall regret very close to the no-teacher baseline.

This behavior is consistent with equation 22: for fixed $K$ and moderate budgets, the upper bound

$$\frac{\mathrm{DynReg}_K}{K} \lesssim K^{-1/3}(\mathcal{V}_{\mathrm{env}} + C)^{1/3} + \frac{C}{K},$$

with $\mathcal{V}_{\mathrm{env}}$ from equation 3, predicts that increasing $C$ by a constant factor should only moderately increase the average regret, especially when the environment variation already dominates. Empirically, we see exactly this regime: the teacher can "spend" up to 20% of the horizon on poisoning without destroying the dynamic-regret guarantees of the base algorithm in Assumption 3.

**(2) Stability is preserved—and often improved in practice.** Perhaps surprisingly, the stability metric $\mathrm{Stab}_K$ defined in equation 11 remains essentially flat as we increase $C_{\mathrm{frac}}$. Across all non-stationarity configurations, the average value of $\mathrm{Stab}_K$ stays around 0.19, and the standard deviation actually shrinks slightly when the budget increases. The worst-case bound in Theorem 1(iii), equation 23, only guarantees that $\mathrm{Stab}_K$ scales with $(\mathcal{V}_{\mathrm{env}} + C)/K$ through the Lipschitz update condition in Assumption 5; it does not rule out the possibility that teaching might *improve* stability by steering the learner toward a fixed target policy. Our experiments show that this benign behavior is typical in the synthetic contextual bandit: the teacher reduces large oscillations by pulling the policy sequence toward $\pi^\dagger$, so the contractive term $\alpha d(\pi_k, \pi_{k-1})$ in equation 16 dominates and smooths the trajectory.

### 4.5 Effect of Environment Non-stationarity

We next examine how the *TRS behavior* changes as we vary the non-stationarity of the environment. Recall that in our generator the total variation budget on the mean rewards is controlled by the number of segments $M$ and the drift scale $\eta$ through Assumption 7 and equation 24: piecewise-stationary instances with larger $M$ and $\eta$ correspond to larger effective variation budgets $B_r$.

For the no-teacher baseline ($C_{\mathrm{frac}} = 0$), we observe that $\mathrm{DynReg}_K$ and $\mathrm{DynReg}_K/K$ remain stable across all $(M, \eta)$, with changes well below 10% even when we move from a stationary single-segment environment ($M = 1$, $\eta = 0$) to highly non-stationary cases ($M = 20$, $\eta = 0.3$). This matches the intuition behind the $K^{2/3}\mathcal{V}_{\mathrm{eff}}^{1/3}$ scaling in equation 12: in the finite-horizon regime we explore, the variation budgets induced by our choices of $(M, \eta)$ are not large enough to dominate the $K^{2/3}$ term, so the regret curves are relatively flat.

When we fix a non-zero budget (e.g., $C_{\mathrm{frac}} = 0.10$) and vary $(M, \eta)$, we again see an essentially stable dynamic regret and stability, while the teaching error $\mathrm{Mismatch}_K$ shows only mild dependence on the drift level. This suggests that, in this regime, the teacher's cost $C$ is the dominant contribution to the effective variation budget $\mathcal{V}_{\mathrm{eff}} \lesssim \mathcal{V}_{\mathrm{env}} + C$ in equation 7: once $C$ is fixed, moderate changes in $\mathcal{V}_{\mathrm{env}}$ do not qualitatively change the behavior.

### 4.6 Summary and Implications

Overall, the synthetic experiments provide a clean empirical picture that is consistent with the simultaneous guarantees suggested by Theorem 1.

- A small poisoning budget $C$ is enough to substantially improve alignment with a target policy $\pi^\dagger$ (small $\mathrm{Mismatch}_K$ in equation 9), while keeping dynamic regret close to that of a strong non-stationary bandit baseline (sublinear $\mathrm{DynReg}_K$ as in equation 22).

- The learner's policy sequence remains stable on average; in fact, the teacher can make the policy *smoother* by suppressing large, purely non-stationary-driven jumps, consistently with the Lipschitz stability guarantee equation 23.

- The qualitative behavior is robust across a range of non-stationarity levels, controlled by $(M, \eta)$ and Assumption 7, indicating that the guarantees of Theorem 1 are not an artifact of a particular environment, but capture a genuine phenomenon in non-stationary RL with poisoning.

From a higher-level perspective, this highlights the importance of our framework: dynamic regret alone (via equation 10) is blind to what policy is being learned, and stability alone (via equation 11) does not prevent convergence to an undesirable policy. By explicitly coupling teaching error, regret, and stability through equation 21–equation 23, we empirically illustrate this coupling in a horizon-one instantiation (a non-stationary contextual bandit), which is a controlled special case of non-stationary RL. This setting isolates the effects of drift and bounded teaching and enables a clear visualization of the achievable TRS profiles obtained by sweeping the budget parameter $C$. Overall, the experiments suggest that joint control of mismatch, regret, and stability is quantitatively achievable with standard algorithms and a simple teacher, providing a first step toward teachable non-stationary RL systems.

## 5 Related Work

Non-stationary RL with variation budgets has been studied in tabular MDPs (Cheung et al., 2020; Fei et al., 2020; Mao et al., 2025), linear and structured settings (Zhou et al., 2022; Feng et al., 2023; Wei et al., 2023; Cheng et al., 2023), and risk-sensitive or constrained formulations (Ding et al., 2023; Wei et al., 2023). Our formulation follows this line by adopting variation budgets on rewards and transitions.

Policy teaching and environment poisoning against RL were formalized in (Rakhsha et al., 2020), which characterized the feasibility and cost of teaching arbitrary target policies by manipulating rewards and transitions. Our work combines such teaching mechanisms with non-stationary RL dynamic-regret bounds.

Finally, our stability condition is inspired by the algorithmic stability literature (Hardt et al., 2016), where Lipschitz-type update rules are used to control generalization error. Here, a similar idea quantifies the smoothness of policy updates in the face of non-stationarity and poisoning.

## 6 Conclusion and Limitations

A dominant theme in non-stationary reinforcement learning is *tracking*: environments drift, algorithms adapt, and success is often summarized by low dynamic regret. This criterion is well aligned with the classical objective of maximizing cumulative reward under drift. At the same time, many deployments introduce additional, application-driven desiderata that are not explicitly represented by dynamic regret alone—for example, alignment to an externally specified reference behavior, and constraints on how abruptly policies may change over time. These considerations do not expose a flaw in the standard formulation; rather, they motivate a different problem setting and a broader evaluation lens when such requirements are present.

In this work we propose such a lens, formalized as the *Teaching–Regret–Stability (TRS) Principle* for *Teachable Non-stationary Reinforcement Learning (TNRL)*. We model a stakeholder (designer/regulator/operator) as a *teacher* who cannot directly replace the learner, but can intervene through bounded modifications of rewards and transitions (environment poisoning) with total cost at most $C$. Within this setting we quantify three objectives: (i) *teaching error* (policy mismatch) with respect to a fixed target policy $\pi^\dagger$, (ii) *dynamic regret* measured in the true drifting environments, and (iii) *policy stability* measured as the step-to-step change of the learned policy sequence. Importantly, our notion of stability concerns the *trajectory of policies* (Eq. equation 11), which is distinct from stability notions defined purely in terms of performance (e.g., stability of rewards or regret).

Our main theorem provides *simultaneous* guarantees showing that, under transparent structural assumptions, there exist teacher–learner pairs for which these three quantities can be jointly controlled: the mismatch can be driven to an arbitrarily small target (up to finite-sample terms), the dynamic regret remains sublinear in the horizon, and the policy sequence evolves smoothly on average. Empirically, in a non-stationary contextual bandit instantiation, we observe that modest teaching budgets can substantially reduce mismatch while leaving regret close to a strong non-stationary baseline and keeping policy stability essentially unchanged.

We view these results as a first step toward principled study of *teachability under drift*: a complementary perspective to regret-based tracking that becomes relevant precisely when stakeholders impose objectives beyond cumulative reward, such as reference-policy alignment and trajectory-level stability.

**Extension to time-varying targets.** Our analysis assumes a stationary reference policy $\pi^\dagger$, which isolates the effect of environmental drift from changes in stakeholder objectives. In applications where the desired reference behavior itself evolves, one can replace $\pi^\dagger$ by a target sequence $\{\pi_k^\dagger\}$ and introduce a *target-variation budget* $\mathcal{V}_\dagger$ (e.g., a cumulative distance $\sum_{k=1}^{K-1} d(\pi_{k+1}^\dagger, \pi_k^\dagger)$ under the same metric $d$ used in the mismatch/stability definitions). Conceptually, the TRS guarantees would then acquire additional terms scaling with $\mathcal{V}_\dagger$, in direct analogy to how non-stationary regret bounds depend on $\mathcal{V}_{\mathrm{env}}$. A full treatment requires choosing the appropriate notion of variation (policy-space vs. induced occupancy) and tracking how it composes with the teacher's budget $C$, which we leave to future work. Future work includes extending the framework to richer forms of non-stationarity, partial observability, and time-varying target policies $\{\pi_k^\dagger\}$ with an explicit target-variation budget, as well as developing learners explicitly optimized for these multi-criteria requirements.

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

# A  Proofs for Lemmas 1–3

## A.1  Auxiliary bounds

**Lemma 6** (Effective variation under poisoning). *For any true environment sequence $\{M_k\}_{k=1}^K$ and any poisoned sequence $\{\tilde{M}_k\}_{k=1}^K$ with per-episode costs $\{c_k\}_{k=1}^K$ defined in equation 4, the effective variation budget equation 6 satisfies*

$$\mathcal{V}_{\text{eff}} \ \leq \ \mathcal{V}_{\text{env}} + 2\sum_{k=1}^K c_k \ = \ \mathcal{V}_{\text{env}} + 2C_{\text{tot}} \ \leq \ \mathcal{V}_{\text{env}} + 2C. \tag{31}$$

*Proof.* We bound the transition-variation part; the reward part is analogous. For any $k \geq 2$ and any $(s, a)$, by the triangle inequality,

$$\|\tilde{P}_k(\cdot|s,a) - \tilde{P}_{k-1}(\cdot|s,a)\|_1 \leq \|\tilde{P}_k(\cdot|s,a) - P_k(\cdot|s,a)\|_1 + \|P_k(\cdot|s,a) - P_{k-1}(\cdot|s,a)\|_1 + \|P_{k-1}(\cdot|s,a) - \tilde{P}_{k-1}(\cdot|s,a)\|_1.$$

Taking $\sup_{s,a}$ on both sides yields

$$\sup_{s,a}\|\tilde{P}_k(\cdot|s,a) - \tilde{P}_{k-1}(\cdot|s,a)\|_1 \leq \sup_{s,a}\|\tilde{P}_k(\cdot|s,a) - P_k(\cdot|s,a)\|_1 + \sup_{s,a}\|P_k(\cdot|s,a) - P_{k-1}(\cdot|s,a)\|_1 + \sup_{s,a}\|\tilde{P}_{k-1}(\cdot|s,a) - P_{k-1}(\cdot|s,a)\|_1.$$

Summing over $k = 2, \ldots, K$ telescopes the poisoning terms and gives

$$\sum_{k=2}^K \sup_{s,a}\|\tilde{P}_k(\cdot|s,a) - \tilde{P}_{k-1}(\cdot|s,a)\|_1 \leq \mathcal{V}_P + 2\sum_{k=1}^K \sup_{s,a}\|\tilde{P}_k(\cdot|s,a) - P_k(\cdot|s,a)\|_1.$$

An identical argument for rewards yields

$$\sum_{k=2}^K \sup_{s,a}|\tilde{r}_k(s,a) - \tilde{r}_{k-1}(s,a)| \leq \mathcal{V}_r + 2\sum_{k=1}^K \sup_{s,a}|\tilde{r}_k(s,a) - r_k(s,a)|.$$

Adding the two inequalities and using the definition of $c_k$ in equation 4 proves equation 31. $\qquad\square$

**Lemma 7** (Per-episode value perturbation (finite-horizon simulation bound)). *Fix an episode $k$ and two MDPs $M_k = (\mathcal{S}, \mathcal{A}, P_k, r_k, \rho_1, \gamma)$ and $\tilde{M}_k = (\mathcal{S}, \mathcal{A}, \tilde{P}_k, \tilde{r}_k, \rho_1, \gamma)$ with the same $(\mathcal{S}, \mathcal{A}, \rho_1, \gamma)$ and horizon $H$. Let $c_k$ be as in equation 4. Then for any policy $\pi$,*

$$\left| V_k(\pi) - \tilde{V}_k(\pi) \right| \leq \kappa_{H,\gamma}\, c_k, \qquad \kappa_{H,\gamma} := \underbrace{\sum_{t=0}^{H-1} \gamma^t}_{=: H_\gamma} + \frac{1}{2} H_\gamma^2. \tag{32}$$

*Proof.* For $h \in \{1, \ldots, H+1\}$, let $V_{k,h}^\pi(s)$ and $\tilde{V}_{k,h}^\pi(s)$ denote the expected discounted return starting from state $s$ at step $h$ when following $\pi$ in $M_k$ and $\tilde{M}_k$, respectively, with the convention $V_{k,H+1}^\pi \equiv \tilde{V}_{k,H+1}^\pi \equiv 0$. Define $\Delta_h := \|V_{k,h}^\pi - \tilde{V}_{k,h}^\pi\|_\infty$.

By the Bellman recursion and the triangle inequality, for any $s$,

$$\left| V_{k,h}^\pi(s) - \tilde{V}_{k,h}^\pi(s) \right| = \left| \mathbb{E}_{a \sim \pi(\cdot|s)} \left[ r_k(s,a) + \gamma\, \mathbb{E}_{s' \sim P_k(\cdot|s,a)} V_{k,h+1}^\pi(s') \right] - \mathbb{E}_{a \sim \pi(\cdot|s)} \left[ \tilde{r}_k(s,a) + \gamma\, \mathbb{E}_{s' \sim \tilde{P}_k(\cdot|s,a)} \tilde{V}_{k,h+1}^\pi(s') \right] \right|$$

$$\leq \sup_{s,a} |r_k(s,a) - \tilde{r}_k(s,a)| + \gamma \sup_{s,a} \left| \mathbb{E}_{P_k(\cdot|s,a)} V_{k,h+1}^\pi - \mathbb{E}_{\tilde{P}_k(\cdot|s,a)} \tilde{V}_{k,h+1}^\pi \right|$$

$$\leq \sup_{s,a} |r_k(s,a) - \tilde{r}_k(s,a)| + \gamma \Delta_{h+1} + \gamma \sup_{s,a} \left| \mathbb{E}_{P_k(\cdot|s,a)} \tilde{V}_{k,h+1}^\pi - \mathbb{E}_{\tilde{P}_k(\cdot|s,a)} \tilde{V}_{k,h+1}^\pi \right|.$$

For the last term, use the standard total-variation inequality: for any bounded $f$ with $\|f\|_\infty \leq B$, $|\mathbb{E}_p f - \mathbb{E}_q f| \leq \frac{B}{2} \|p - q\|_1$. Since rewards lie in $[0, 1]$, we have $\|\tilde{V}_{k,h+1}^\pi\|_\infty \leq H_\gamma$. Thus,

$$\sup_{s,a} \left| \mathbb{E}_{P_k(\cdot|s,a)} \tilde{V}_{k,h+1}^\pi - \mathbb{E}_{\tilde{P}_k(\cdot|s,a)} \tilde{V}_{k,h+1}^\pi \right| \leq \frac{H_\gamma}{2} \sup_{s,a} \|P_k(\cdot|s,a) - \tilde{P}_k(\cdot|s,a)\|_1.$$

Combining the bounds gives the recursion

$$\Delta_h \leq \sup_{s,a} |r_k(s,a) - \tilde{r}_k(s,a)| + \gamma \Delta_{h+1} + \gamma \frac{H_\gamma}{2} \sup_{s,a} \|P_k(\cdot|s,a) - \tilde{P}_k(\cdot|s,a)\|_1.$$

Unrolling from $h = 1$ to $H$ and using $\Delta_{H+1} = 0$ yields

$$\Delta_1 \leq H_\gamma \sup_{s,a} |r_k(s,a) - \tilde{r}_k(s,a)| + \frac{H_\gamma^2}{2} \sup_{s,a} \|P_k(\cdot|s,a) - \tilde{P}_k(\cdot|s,a)\|_1.$$

Finally, $|V_k(\pi) - \tilde{V}_k(\pi)| \leq \Delta_1$ (values start from $\rho_1$), and $c_k$ is the sum of the two sup terms in equation 4, which proves equation 32. □

## A.2 Proof of Lemma 1

*Proof of Lemma 1.* Fix $\varepsilon > 0$ and assume $C \geq C_{\min}(\varepsilon)$. By Assumption 4(iii), the teacher may choose $\tilde{M}_k \equiv M^c(\varepsilon)$ for all $k$, which satisfies $C_{\text{tot}} \leq C$.

Since the learner interacts with the fixed canonical MDP $M^c(\varepsilon)$ for $K$ episodes, Assumption 4(ii) gives

$$\frac{1}{K} \sum_{k=1}^{K} \mathbb{E}\big[ V^c(\pi^\dagger) - V^c(\pi_k) \big] \leq G_c \varepsilon + \tilde{\mathcal{O}}\Big(\frac{1}{K}\Big).$$

Applying the coupling inequality in Assumption 4(i), $V^c(\pi^\dagger) - V^c(\pi_k) \geq G_c d(\pi_k, \pi^\dagger)$, and averaging over $k$ yields

$$\text{Mismatch}_K = \frac{1}{K} \sum_{k=1}^{K} \mathbb{E}\big[ d(\pi_k, \pi^\dagger) \big] \leq \varepsilon + \tilde{\mathcal{O}}\Big(\frac{1}{K}\Big),$$

which is exactly equation 17. □

### A.3 Proof of Lemma 2

*Proof of Lemma 2.* By Assumption 3, for any poisoned environment sequence with effective variation budget $\mathcal{V}_{\text{eff}}$,

$$\text{DynReg}_K^{(\tilde{M})} \leq \tilde{\mathcal{O}}\big(K^{2/3}\mathcal{V}_{\text{eff}}^{1/3}\big).$$

By Lemma 6, $\mathcal{V}_{\text{eff}} \leq \mathcal{V}_{\text{env}} + 2C_{\text{tot}} \leq \mathcal{V}_{\text{env}} + 2C$. Plugging this into the dynamic-regret bound yields

$$\text{DynReg}_K^{(\tilde{M})} \leq \tilde{\mathcal{O}}\Big(K^{2/3}(\mathcal{V}_{\text{env}} + 2C)^{1/3}\Big) = \tilde{\mathcal{O}}\Big(K^{2/3}(\mathcal{V}_{\text{env}} + C)^{1/3}\Big),$$

where the last step absorbs the constant factor 2 into the $\tilde{\mathcal{O}}(\cdot)$ notation. This proves equation 18. $\qquad\square$

### A.4 Proof of Lemma 3

*Proof of Lemma 3.* For each episode $k$, let $\tilde{V}_k(\pi)$ denote the value of policy $\pi$ in the poisoned MDP $\tilde{M}_k$. Let $\tilde{\pi}_k^\star \in \arg\max_\pi \tilde{V}_k(\pi)$ be an optimal policy in $\tilde{M}_k$. Then

$$\text{DynReg}_K = \sum_{k=1}^{K} \Big(V_k(\pi_k^\star) - V_k(\pi_k)\Big)$$

$$= \sum_{k=1}^{K} \Big(\underbrace{V_k(\pi_k^\star) - \tilde{V}_k(\pi_k^\star)}_{(\text{I})} + \underbrace{\tilde{V}_k(\pi_k^\star) - \tilde{V}_k(\pi_k)}_{(\text{II})} + \underbrace{\tilde{V}_k(\pi_k) - V_k(\pi_k)}_{(\text{III})}\Big).$$

We bound the three terms.

**Step 1: terms (I) and (III) via value perturbation.** By Lemma 7, for any policy $\pi$, $|V_k(\pi) - \tilde{V}_k(\pi)| \leq \kappa_{H,\gamma} c_k$. Applying this to $\pi = \pi_k^\star$ and $\pi = \pi_k$ yields

$$(\text{I}) + (\text{III}) \leq 2\kappa_{H,\gamma} c_k.$$

**Step 2: term (II) is upper bounded by the poisoned regret.** Since $\tilde{\pi}_k^\star$ is optimal for $\tilde{V}_k$, we have $\tilde{V}_k(\pi_k^\star) \leq \tilde{V}_k(\tilde{\pi}_k^\star)$, hence

$$(\text{II}) = \tilde{V}_k(\pi_k^\star) - \tilde{V}_k(\pi_k) \leq \tilde{V}_k(\tilde{\pi}_k^\star) - \tilde{V}_k(\pi_k).$$

Summing over $k$ gives

$$\sum_{k=1}^{K}(\text{II}) \leq \sum_{k=1}^{K} \Big(\tilde{V}_k(\tilde{\pi}_k^\star) - \tilde{V}_k(\pi_k)\Big) = \text{DynReg}_K^{(\tilde{M})}.$$

**Step 3: combine and use the budget constraint.** Putting the bounds together,

$$\text{DynReg}_K \leq \text{DynReg}_K^{(\tilde{M})} + 2\kappa_{H,\gamma}\sum_{k=1}^{K} c_k \leq \text{DynReg}_K^{(\tilde{M})} + 2\kappa_{H,\gamma}C_{\text{tot}} \leq \text{DynReg}_K^{(\tilde{M})} + 2\kappa_{H,\gamma}C.$$

This is equation 19 with the constant in $\mathcal{O}(C)$ equal to $2\kappa_{H,\gamma}$. $\qquad\square$

## B Additional Theoretical Results

### B.1 Proof of Lemma 4

We provide a complete proof of the stability bound stated in Lemma 4.

*Proof of Lemma 4.* For brevity, write

$$d_k := d(\pi_k, \pi_{k-1}), \qquad \Delta_k^P := \sup_{s,a} \big\|\tilde{P}_k(\cdot \mid s,a) - \tilde{P}_{k-1}(\cdot \mid s,a)\big\|_1,$$

$$\Delta_k^r := \sup_{s,a} \big| \tilde{r}_k(s,a) - \tilde{r}_{k-1}(s,a) \big|.$$

By Assumption 5, for all $k \geq 3$ we have

$$d_k \ \leq \ \alpha \, d_{k-1} + L_P \Delta_k^P + L_r \Delta_k^r. \tag{33}$$

**Step 1: Unrolling the recursion.** We first show by induction that for all $k \geq 3$,

$$d_k \ \leq \ \alpha^{k-2} d_2 + \sum_{j=2}^{k} \alpha^{k-j} \big( L_P \Delta_j^P + L_r \Delta_j^r \big). \tag{34}$$

For $k = 3$, equation 33 gives
$$d_3 \leq \alpha d_2 + L_P \Delta_3^P + L_r \Delta_3^r,$$
which coincides with equation 34 for $k = 3$. Assume equation 34 holds for some $k \geq 3$. Then, using equation 33,

$$d_{k+1} \leq \alpha d_k + L_P \Delta_{k+1}^P + L_r \Delta_{k+1}^r$$

$$\leq \alpha \Big( \alpha^{k-2} d_2 + \sum_{j=3}^{k} \alpha^{k-j} (L_P \Delta_j^P + L_r \Delta_j^r) \Big) + L_P \Delta_{k+1}^P + L_r \Delta_{k+1}^r$$

$$= \alpha^{k-1} d_2 + \sum_{j=3}^{k} \alpha^{(k+1)-j} (L_P \Delta_j^P + L_r \Delta_j^r) + \alpha^0 (L_P \Delta_{k+1}^P + L_r \Delta_{k+1}^r)$$

$$= \alpha^{(k+1)-2} d_2 + \sum_{j=3}^{k+1} \alpha^{(k+1)-j} (L_P \Delta_j^P + L_r \Delta_j^r),$$

which is exactly equation 34 with $k$ replaced by $k + 1$. Thus equation 34 holds for all $k \geq 3$.

**Step 2: Bounding the average policy change.** By definition,

$$\mathrm{Stab}_K = \frac{1}{K-1} \sum_{k=2}^{K} d_k = \frac{1}{K-1} \Big( d_2 + \sum_{k=3}^{K} d_k \Big).$$

Using equation 34 for $k \geq 3$, we obtain

$$\sum_{k=2}^{K} d_k \leq d_2 + \sum_{k=3}^{K} \Big( \alpha^{k-2} d_2 + \sum_{j=2}^{k} \alpha^{k-j} (L_P \Delta_j^P + L_r \Delta_j^r) \Big)$$

$$= d_2 \sum_{k=2}^{K} \alpha^{k-2} + \sum_{k=3}^{K} \sum_{j=2}^{k} \alpha^{k-j} (L_P \Delta_j^P + L_r \Delta_j^r).$$

We bound the two terms separately. For the first term, since $\alpha \in [0,1)$, the geometric series is bounded:

$$\sum_{k=2}^{K} \alpha^{k-2} = \sum_{m=0}^{K-2} \alpha^m \leq \frac{1}{1-\alpha},$$

so

$$\frac{1}{K-1} d_2 \sum_{k=2}^{K} \alpha^{k-2} \leq \frac{d_2}{(1-\alpha)(K-1)} = \mathcal{O}\Big( \frac{1}{K} \Big).$$

For the second term, define

$$S := \sum_{k=3}^{K} \sum_{j=2}^{k} \alpha^{k-j} (L_P \Delta_j^P + L_r \Delta_j^r).$$

We exchange the order of summation:

$$S = \sum_{j=2}^{K}(L_P\Delta_j^P + L_r\Delta_j^r)\sum_{k=j}^{K}\alpha^{k-j}.$$

Again using $\alpha \in [0,1)$ and the geometric series,

$$\sum_{k=j}^{K}\alpha^{k-j} = \sum_{m=0}^{K-j}\alpha^m \le \frac{1}{1-\alpha},$$

hence

$$S \le \frac{1}{1-\alpha}\sum_{j=2}^{K}(L_P\Delta_j^P + L_r\Delta_j^r).$$

Therefore

$$\mathrm{Stab}_K \le \frac{1}{K-1}\cdot\frac{1}{1-\alpha}\sum_{j=2}^{K}(L_P\Delta_j^P + L_r\Delta_j^r) + \mathcal{O}\Big(\frac{1}{K}\Big).$$

Using $L_P\Delta_j^P + L_r\Delta_j^r \le (L_P + L_r)(\Delta_j^P + \Delta_j^r)$ and the fact that, up to universal constants,

$$\sum_{j=2}^{K}(\Delta_j^P + \Delta_j^r) \lesssim \mathcal{V}_{\mathrm{env}} + C$$

(Assumptions 1 and 2), we obtain

$$\mathrm{Stab}_K \le \frac{L_P + L_r}{1-\alpha}\cdot\frac{\mathcal{V}_{\mathrm{env}} + C}{K-1} + \mathcal{O}\Big(\frac{1}{K}\Big),$$

which is equivalent to equation 20 after replacing $K-1$ by $K$ in the denominator. $\square$

**Remark 3.** *The parameter $\alpha \in [0,1)$ in Assumption 5 plays the role of a contraction factor for the policy update: when the environment is fixed ($\Delta_k^P = \Delta_k^r = 0$), the recursion $d_k \le \alpha d_{k-1}$ implies that successive policy changes decay at rate $\alpha^k$. The bound $\sum_{k=j}^{K}\alpha^{k-j} \le (1-\alpha)^{-1}$ used above is the standard geometric-series estimate associated with this contraction.*

### B.2 Proof of Lemma 5

*Proof.* Fix $t$ and $(x,a)$. Then

$$
\begin{aligned}
\big|\bar{r}_{t+1}(x,a) - \bar{r}_t(x,a)\big| &= \big|\sigma(\langle\theta_{t+1},\phi(x,a)\rangle) - \sigma(\langle\theta_t,\phi(x,a)\rangle)\big| \\
&\le L_\sigma\big|\langle\theta_{t+1} - \theta_t,\phi(x,a)\rangle\big| \quad \text{(by Lipschitzness of } \sigma) \\
&\le L_\sigma\|\theta_{t+1} - \theta_t\|_2\|\phi(x,a)\|_2 \quad \text{(Cauchy–Schwarz)} \\
&\le L_\sigma L_\phi\|\theta_{t+1} - \theta_t\|_2,
\end{aligned}
$$

where we used the uniform bound on $\|\phi(x,a)\|_2$ in the last step. Taking the supremum over $(x,a)$ yields

$$\big\|\bar{r}_{t+1} - \bar{r}_t\big\|_\infty \le L_\sigma L_\phi\|\theta_{t+1} - \theta_t\|_2.$$

Summing over $t = 1,\ldots,T-1$ gives

$$\sum_{t=1}^{T-1}\big\|\bar{r}_{t+1} - \bar{r}_t\big\|_\infty \le L_\sigma L_\phi\sum_{t=1}^{T-1}\|\theta_{t+1} - \theta_t\|_2 = L_\sigma L_\phi B_\theta,$$

which is exactly equation 28. $\square$

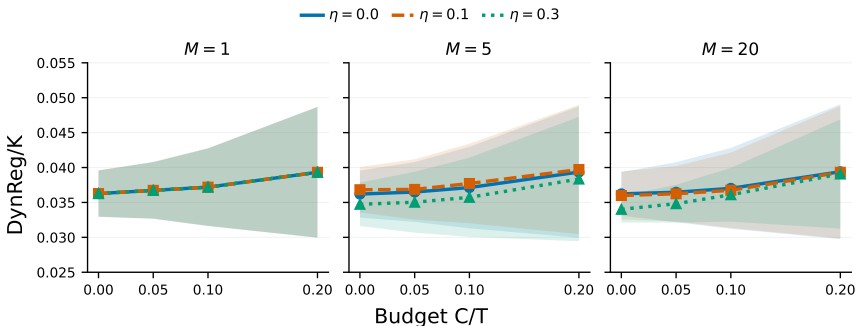

Figure 1: Average dynamic regret $\text{DynReg}_K/K$ versus normalized poisoning cost per step $C/T$. Each curve corresponds to a different non-stationarity configuration $(M, \eta)$; shaded regions denote standard deviation over seeds.

### B.3 Expected path length of the synthetic generator

**Lemma 8** (Expected path length of the synthetic generator). *In Algorithm 1, let $B_\theta$ be defined as in equation 27. Then there exists a constant $c_d > 0$ depending only on the dimension $d$ such that $\mathbb{E}[B_\theta] \leq c_d M \eta$.*

### B.4 Evaluation Metrics

We report three metrics that correspond directly to the quantities in Theorem 1.

**Teaching error (policy mismatch).** We measure how well the teacher succeeds at steering the learner towards the fixed target policy $\pi^\dagger$ by the average mismatch

$$\text{Mismatch}_K = \frac{1}{K} \sum_{k=1}^{K} \mathbb{E}\big[d(\pi_k, \pi^\dagger)\big],$$

where $d(\cdot, \cdot)$ is the $\ell_1$ distance between action distributions, taken uniformly over contexts.

**Dynamic regret in the true environment.** Although all updates happen in the (possibly poisoned) environments, performance is evaluated in the true non-stationary environment. We therefore track the cumulative dynamic regret

$$\text{DynReg}_K = \sum_{k=1}^{K} \big(V_k(\pi_k^\star) - V_k(\pi_k)\big),$$

where $V_k(\pi)$ denotes the expected return of policy $\pi$ in the true environment at episode $k$, and $\pi_k^\star$ is the per-episode optimal policy. We report both the cumulative regret and the average regret $\text{DynReg}_K/K$.

**Policy stability.** Finally, we quantify the smoothness of the policy trajectory via the average step-to-step change

$$\text{Stab}_K = \frac{1}{K-1} \sum_{k=2}^{K} \mathbb{E}\big[d(\pi_k, \pi_{k-1})\big].$$

Small values of $\text{Stab}_K$ indicate that the learner updates its policy gradually over time, whereas large values point to unstable behavior with frequent drastic shifts.

## C Additional Experimental Figures

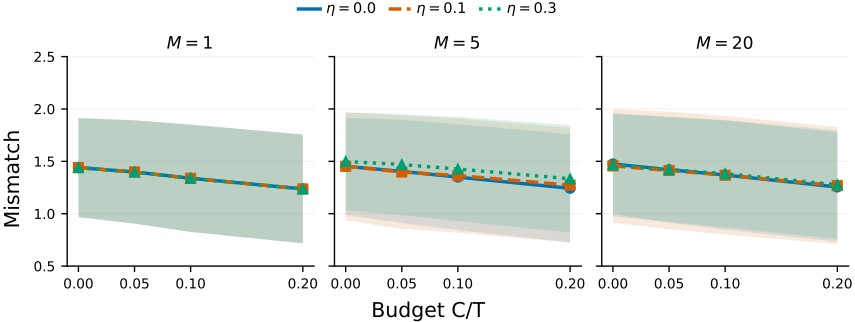

Figure 2: Teaching error Mismatch$_K$ versus normalized poisoning cost per step $C/T$. Larger budgets consistently reduce policy mismatch across all non-stationarity levels; shaded regions denote standard deviation over seeds.

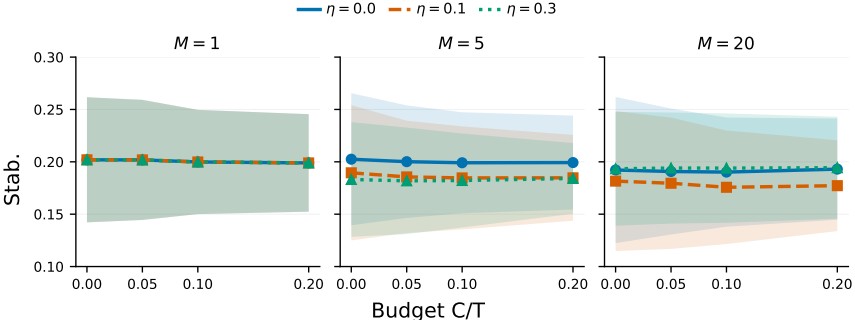

Figure 3: Policy stability Stab$_K$ versus normalized poisoning cost per step $C/T$. Stability remains essentially flat, indicating that teaching does not destabilize the policy updates; shaded regions denote standard deviation over seeds.

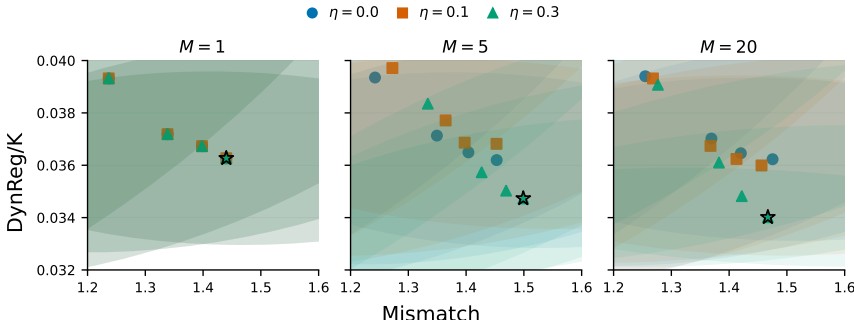

Figure 4: Regret–mismatch frontier for each $M$. Each point is the mean over seeds of (Mismatch, DynReg/$K$) for a specific $(\eta, C/T)$; marker shapes indicate $\eta$ and colors match the $\eta$ legend. A translucent ellipse shows the empirical covariance ($1\sigma$) of seed-level outcomes. The starred marker highlights the minimum mean dynamic regret within each panel.

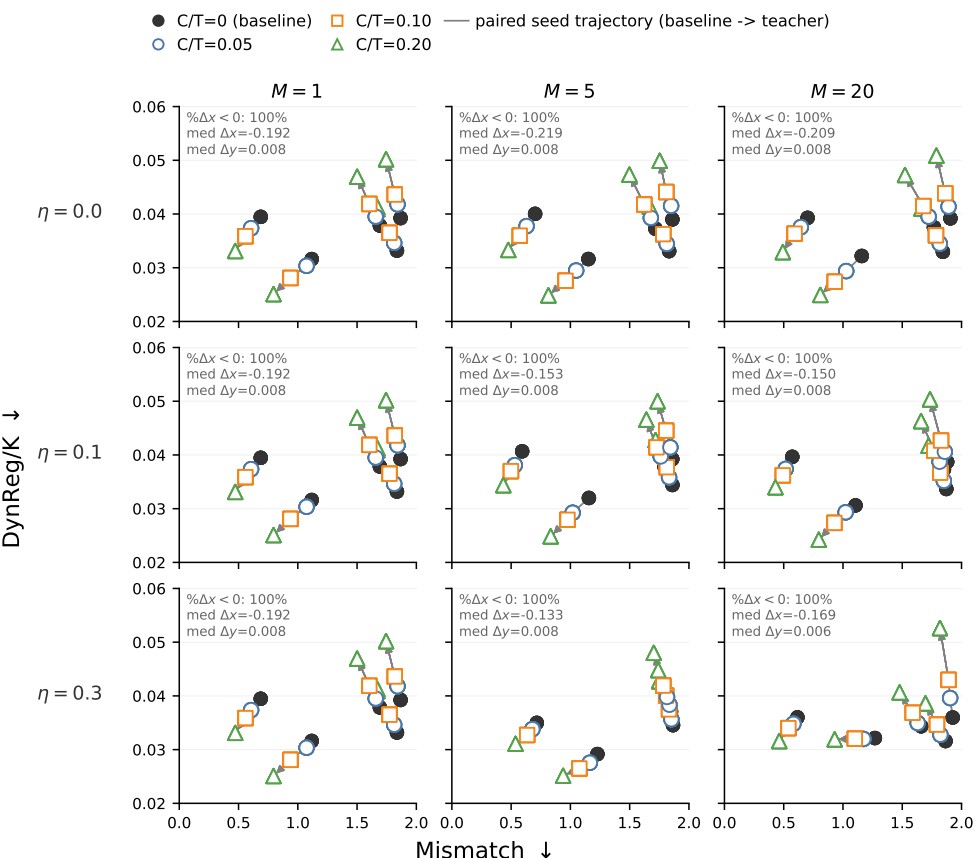

Figure 5: Paired seed trajectories in mismatch–dynamic regret space for each $(\eta, M)$. Gray arrows connect the same seed from the baseline $(C/T = 0)$ through increasing budgets, with arrowheads at the largest budget. Hollow markers denote teacher budgets $(C/T \in \{0.05, 0.10, 0.20\})$, and filled dots denote the baseline.

