# OpenReview forum: "The Teaching--Regret--Stability Principle in Non-Stationary Reinforcement Learning"
_TMLR — Withdrawn by Authors_

### Review · Reviewer_HTLy · 2026-01-08

**Summary Of Contributions:**

__(1) Summary.__ This paper studies the problem of reinforcement learning (RL) in non-stationary, episodic settings, with an focus on better understanding how (possibly competing) objectives of teachability, stability, and regret should interplay with one another. With this focus in mind, the starting point for the work questions whether two common assumptions belong in the study of non-stationary RL: first, that our only objective should be low-regret, and second, instability is a tolerable side-effect of how most methods work. The paper proposes reconsidering these two assumptions, arguing that the _teachability_ of an algorithm and its _stability_ should count as first class characteristics that we scrutinize when contrasting different methods. With this framing in mind, the paper's core technical contribution is a series of definitions and assumptions that build toward the main result: Theorem 1. This theorem codifies the "Teaching–Regret–Stability (TRS) Principle", which advocates for the simultaneous satisfaction of all three properties (1: teachability, 2: low regret, and 3: stability). The technical work is then to set out the needed groundwork in order to prove this Theorem in a suitably general setting.

The setting considered is a somewhat simplified variant of non-stationary RL in which a learning algorithm interacts with an MDP across K episodes. Each episode, the MDP is allowed to shift by a bounded amount, and the algorithm plays a fixed policy for the duration of the episode.  The TRS principle includes three key measures: (1) the teaching error, quantifying the mismatch between the algorithm's played policy and the teacher's desired policy $\pi^\dagger$; (2) the (dynamic) regret of the algorithm in the true environment (not the shaped/poisoned one presented by the teacher); and (3) the stability of the policy sequence played by the learner.

Theorem 1 proves that there is a teaching strategy which, subject to suitable assumptions, ensures all three measures are bounded in a non-vacuous way.

In additional to the theoretical groundwork laid that builds to Theorem 1 (and of course the theorem itself), the paper also presents an experimental study using non-stationary contextual bandits to further examine the developed theory. The results reported summarize the teaching error, dynamic regret, and stability for different poisoning budgets.

__(2) Strengths and Weaknesses.__

[Strengths]
The paper possesses several strengths. It is clear in its objectives, and clear in its conceptual and formal developments. Each piece of notation is carefully defined. The formalism and motivation were easy to follow throughout, and the main result is non-trivial.

[Weaknesses]
The biggest weakness of the work is in its framing. There are a few key aspects of the framing that I have a hard time motivating in the paper's current form:
1. The teacher wants the agent to reach a pre-determined policy: There are two issues here. First, if the teacher knows $\pi_\dagger$, why would they bother trying to teach it to a learner? Second, if the world is truly non-stationary, why would the desired solution be a fixed, stationary policy?
2. Reward, regret, and teaching: A related point, but under the reward hypothesis, the maximization of (some suitable function of) reward is the goal. Thus, in the start of the work, it is stated: "... without asking what policy it is converging to, or whether that policy aligns with
any external goal beyond cumulative reward." Under the reward hypothesis, the external goal _is_ the maximization of the reward. Now, it is fair to reject the reward hypothesis and instead opt for an alternative formulation of goals: here, for instance, the paper seems to advocate for a multi-criteria or multi-objective approach in which we want policies that live along the frontier of maximizing (1) stability, (2) reward, and (3) teachability. That's perfectly fine, but I believe this fact would be better articulated explicitly, rather than implicitly. That is: the introduction should make it explicit that it is pushing back on the reward hypothesis and/or embracing a multi-criteria approach.
3. Trade-off vs. Multi-criteria. Third, the supporting text of the main theorem suggests that it reveals a trade-off. I do not perceive it this way. The theorem establishes that three upper bounds can simultaneously hold. It does not show that one of the three quantities comes at the expense of the other (further supported by Table 1: stability is unaffected).
4. Lastly, I find the setting to restrictive for studying non-stationarity. I can appreciate for deep technical results that restricting the setting is valuable.

Despite this, the main result is true, and I believe is worth sharing with the community.

**Audience:**

Yes

**Audience Explanation:**

The paper studies a popular problem in RL, and gives a new result regarding the relationship between teachability, regret, and stability.

**Claims And Evidence:**

Yes

**Claims Explanation:**

Yes, with the exception of the use of the term "trade-off" used in the discussion of the TRS principle. For instance, "We interpret this as a three-dimensional trade-off frontier—the TRS Principle—that any teachable nonstationary RL system must inhabit.". As discussed above, I do not believe Theorem 1 implies a trade-off of these quantities on its own, but rather that three inequalities can simultaneously be satisfied. To establish a trade-off, we would further require that changes to one quantity necessarily change the others. I do not believe this is proven in the present paper, so would advise changing this language.

Otherwise, all theoretical results are proven with detailed proofs given in the Appendix.

**Requested Changes:**

- I would change the title. I do not believe "Regret Is Not Enough" communicates the right aspects of the work. The subtitle "Teaching and Stability in Non-Stationary Reinforcement Learning" is better. I would use "The Teaching-Regret-Stability Principle in Non-Stationary Reinforcement Learning".
- Why can't $\gamma = 0$? Especially since you run experiments with contextual bandits, I would suggest allowing $\gamma \in [0,1]$.
- I would suggest adding further exposition motivating why we might care about settings where the world is non-stationary, but a teacher picks a fixed, stationary policy as the target. This feels like a bizarre objective (rather than a teacher that picks a non-stationary policy, or picks a new stationary policy every time the environment shifts).
- I would expand on the treatment of reward, stability, and teachability under the framing of going beyond the reward hypothesis (see, for instance, Settling the Reward Hypothesis by Bowling et al., 2023). It is fair of course to value more than just scalar reward, but I believe the argument will be much sharper if it is made explicit rather than implicit. Otherwise there is conceptual confusion about what the goal is: if we agree to the reward hypothesis, then we must also further agree that maximizing reward is the satisfaction of external goals (and therefore things like stability/teachability must be reflected in the reward).

---

### Review · Reviewer_hD9F · 2026-01-16

**Summary Of Contributions:**

This paper introduces a framework called Teachable Non-stationary RL (TNRL) and the Teaching–Regret–Stability (TRS) Principle. Central to the paper's contribution is that dynamic regret alone is an insufficient metric for evaluating non-stationary RL algorithms, particularly in safety-critical domains where stakeholders have target policies in mind. The authors provide a formalization building where a "teacher" can  manipulate environments (with bounded variation) to steer a learning algorithm toward a target policy. They prove a main theorem (Theorem 1) showing that under certain assumptions, one can simultaneously achieve low teaching error (guiding the policy to the target), sublinear dynamic regret in the true environment, and stable policy updates (measured by the sup norm over the change in policy). Empirically, a synthetic non-stationary contextual bandit is used to validate the trade-off between teaching error, regret and stability.

### Strengths
1. Novel problem formulation pursuing a seemingly novel question: can non-stationary RL algorithms be "taught" to converge to a policy to satisfy external desiderata while maintaining performance guarantees? This perspective bridges environment poisoning, dynamic regret, and stability, offering a potentially useful lens for thinking about RL systems that must satisfy stakeholder constraints beyond reward maximization.

2. Clean theoretical framework with interpretable bounds: The main contribution is a set of bounds (Equations 19–21) that relate "teaching", regret and stability. The Lipschitz policy-update assumption (Assumption 5) is reasonable and the stability analysis (Lemma 4) appears sound. Theorem 1 combines these results and aims to quantify the trade-off between teaching, regret, and stability.

### Weaknesses
1. The theoretical contribution appear limited: The main theorem (Theorem 1) follows directly from concatenating Lemmas 1–4, most of which rely on standard arguments or are claimed to follow from prior work. Lemma 1 appeals to "standard arguments in the environment poisoning literature" without proof; Lemma 2 is a direct consequence of assumed dynamic regret bounds; Lemma 3 uses standard MDP perturbation arguments. Only Lemma 4 receives a complete proof (in the appendix). For a theory-focused submission, the lack of formal proofs for the key lemmas makes the contribution feel incomplete. If these results are indeed straightforward applications of existing techniques, the paper's theoretical novelty is diminished.

2. The "TRS Principle" is never formally stated, and the claimed trade-off is unclear: Despite being the paper's central contribution, the TRS Principle is discussed only informally. What exactly is being traded off? By Equation 19, teaching error appears independent of the budget C (it depends on ε and K, which are determined a priori). Equations 20–21 show that both regret and stability depend on C in the same direction (increasing C worsens both bounds). This suggests the relationships are more like constraints than a frontier with a meaningful trade-off. The paper would benefit from a formal statement of the principle and clearer articulation of what parameters govern the trade-off.

3. The experimental scope: Much of the paper (Sections 1-3) are positioned as addressing non-stationary reinforcement learning, yet the experiments (Section 4) are entirely on contextual bandits. While bandits are a useful simplification, the claim that "we obtain a genuinely three-dimensional view of non-stationary RL" (Section 4.6) is overclaiming given this limited scope. Additionally, the teacher's poisoning strategy to guide the learner toward the canonical environment is never explicitly stated (Section 4.2), and the appendix figures show confidence intervals that span nearly the entire y-axis range, raising questions about statistical significance of the observed differences.

4. The paper's framing overstates the limitations of existing work: The introduction and conclusion characterize standard non-stationary RL as having a "disaster scenario" or "structural failure mode," but this framing is not well supported:

   "instability is a technical artifact" (Section 1) is not well motivated. Would a low dynamic regret algorithm not necessarily exhibit some degree of stability in performance? The notion of stability used in the paper (policy stability, Equation 11) , may differ from other natural meansures "stability" in the regret-minimization context (stability in performance, for example). However, this distinction is not clarified.

   "in real deployments, there is almost always a designer... who has a target policy in mind" is interesting, but it acknowledges a fundamentally different problem setting where the objective includes matching a target policy, rather than exposing a flaw in existing formulations.

   "[if a learner is not teachable while preserving low dynamic regret and maintaining stability then] low dynamic regret is a dangerously incomplete certificate" does not follow logically. An algorithm could be excellent at tracking changing optima while being difficult to steer toward an arbitrary target; these are separate properties.

   "disaster scenario" where two algorithms have identical regret but one is "catastrophically misaligned" appears to be simply the observation that regret does not measure alignment with external desiderata. I do not know of any similar claims that have been made in the literature. The appropriate response is to measure what one cares about, not to characterize regret-based evaluation as dangerously incomplete.

   The proposed problem is interesting, but suggesting that existing non-stationary RL "overlooks" stakeholder concerns or has hidden failure modes is overstated. The standard formulation optimizes cumulative reward by definition; if one has additional objectives (target policy alignment, specific stability notions), these constitute a different problem that may warrant different metrics.

**Audience:**

Yes

**Audience Explanation:**

There is substantial interest in continual learning in many subcommunities at present. This ranges from better empirical practice, to devising better algorithms, as well as better theoretical characterizations of the problem (including various notions of regret). This paper's framework is relatively clear and appears novel. Despite some disconnect between the contributions and the evidence (see above), there would be an audience for a clear articulation of a trade-off potentially relevant to real world scenarios.

**Claims And Evidence:**

No

**Claims Explanation:**

There are a few shortcomings of this paper which contribute to me leaning towards "no". For instance, the rhetorical framing around the problem setting is not an accurate characterization of the contribution. The theoretical contributions and the experiments are lacking some details which make the contributions less convincing. Finally, the discussion around the "TRS principle" is not clear.

**Requested Changes:**

Critical:
- Provide complete proofs for Lemmas 1–3 in the appendix, or clearly explain which existing results they follow from and how.
- Formally define the TRS Principle and clarify what constitutes the "trade-off" given that the bounds in Equations 19–21 do not obviously describe a Pareto frontier.
- Explicitly describe the teacher's poisoning strategy used in experiments (Section 4.2).

Would strengthen the paper:
- Revise the rhetorical framing to present TNRL as a complementary problem formulation rather than as exposing fundamental flaws in existing work. The current framing risks alienating readers who may otherwise find the technical contributions interesting.
- Include experiments on tabular MDPs or other RL settings beyond contextual bandits to support the general RL framing. Currently, the claim that "we obtain a genuinely three-dimensional view of non-stationary RL" (Section 4.6) is overclaiming given that bandits are a horizon-one special case with no transitions.
- Address the large confidence intervals in Figures 2–4 (appendix), which span nearly the entire y-axis range and raise questions about the statistical significance of observed differences.
- Discuss whether the assumption of known variation budget (Assumption 1) can be relaxed, following recent work on parameter-free dynamic regret algorithms.

---

### Review · Reviewer_6hHH · 2026-02-24

**Summary Of Contributions:**

The paper studies the problem of teaching a target policy in an MDP in a non-stationary environment: assuming the learner implements a policy with dynamic regret guarantees and small per-step changes, it is shown -- under some assumptions -- how these guarantees change if a teacher modifies the environment to teach the target policy, and how effective the latter teaching strategy is.

I have several criticisms regarding the manuscript, which prevent me from recommending acceptance.

**Problem setting:** I find the problem setting largely unmotivated. The second paragraph in the introduction describes very generally when this situation might happen, but for me it would be helpful to see a concrete instantiation. In particular, it is not clear where the constraints are coming from. If a regulator's aim is to ensure a certain target policy is implemented and has the ability to change the environment in every step, why can't they simply change it to the one where the target policy is optimal, and why do they need to satisfy some constraints?

**Approach:** The paper makes several assumptions under which it shows some trade-off between poisoning (how much the teacher can change the environment) and the learner's behavior (dynamic regret, average change of the learner's policy, called "stability"). These assumptions give an example setting when achieving the targeted trade-off is possible (I also find some of the assumptions unmotivated; details are provided below). Instead of this approach, it would be much more insightful to formalize when the trade-off is achievable (on top of rephrasing the statements, on the technical side this would require showing lower bounds).

**Technical content:** I find the technical content of the paper quite shallow, as the proofs mostly rely on the triangle inequality and straightforward recursions. This means that the main technical content should be in the assumptions which make the proofs straightforward, but these are largely collected from existing results. The notable difference is Assumption 4, which captures the feasibility of the problem: the existence of an MDP where the target policy is learnable and which is not too far from the real environment. The paper defines this distance as $C_{min}$, and treats the resulting value as some kind of an optimum, but this is not guaranteed at all. In my view the proper definition would be to find an MDP $M^c$ which minimizes $C_{min}$ under the constraints formulated by the other parts of the assumptions. This would be a step towards my suggestion in the previous paragraph -- replacing Assumption 4 with a computable quantity which measures the problem hardness, and also strengthening the statement of Theorem 1 and eliminating some incorrect statements (e.g., at the beginning of Section 3 and before Theorem 1, it is said that $\epsilon(C)$ is non-increasing, but this is not guaranteed because of the arbitrary choice of $M^c$).

**Experiments:** While the average values computed in the experiments indicate some kind of expected behavior, the confidence intervals completely overlap, so no claims in the section are actually supported by the experimental results. The paper also makes claims about trade-offs and dominating terms, but these are not analyzed by varying different parameters... Also, it is not possible to reproduce any of the experiments from the descriptions (providing the code is not a substitute for this). Furthermore, even if some results are relegated to the appendix, they should be referenced in the main text.

**Technical errors / other comments**
- On page 2, the description of Theorem 1 is overstated: The dynamic regret bound contains a term $C$ which is not necessarily sublinear in the time horizon. In fact, the way the environment poisoning is introduced in Algorithm 2, the poisoning budget is $\lambda T$, which is linear in the time horizon.
- The reward function is noiseless in Sections 2 and 3, and noisy in Section 4, creating a mismatch in the settings (also, the state space is finite in the first part, and infinite in the second).
- The relation $\lesssim$ is undefined (equation 7).
- Instead of Assumptions 1 and 3 (especially the communicating assumption), one could assume that the learner can achieve a regret bound $R(K,V_{eff} )$, and this bound would then become essentially $R(K, V_{eff} + C)$ in the main theorem, providing a cleaner black-box reduction. Make the last paragraph of Assumption 3 a proper sentence. The assumption states that the bound can be achieved without prior knowledge, but no reference is provided (e.g., C-Y Wei and H. Luo, Non-stationary reinforcement learning without prior knowledge: an optimal black-box approach, COLT, 2021). The reference provided for the bandit case requires prior knowledge of $V_{eff}$ and $T$.
- Assumption 4: Why is there the same constant $G_c$ in both parts (i) and (ii)? Part (i) requires that $\pi^\dagger$ reaches every state with positive probability (otherwise $\pi$ and $\pi^\dagger$ could be different in a state with 0 probability without changing the value function -- or the distance could be redefined to only consider states with positive occupation measure under the policies). I have already commented about the "non-minimality" of $C_{min}$ above.
- Assumption 5: Which algorithm satisfying Assumption 3 satisfies the Lipschitz policy update Assumption 5? The algorithms I am aware of either require prior knowledge or reset the policies from time-to-time.
- What is *robust optimal* below equation (17)?
- Reward definition in the experiments: In Assumption 6, if $r_t \in [0,1]$, then it is automatically sub-Gaussian. The constant $\sigma$ is later used as the (undefined) squashing function. The clipped reward in Algorithm 1 does not generally have the correct expectation $\bar{r}$.
- How is it supported, for example, that "poisoning *significantly* improves how closely the learner tracks the target policy" where all results are within statistical variations?
- Algorithm 2: Is the whole $\tilde{r}_t$ revealed to the learner, or just the one corresponding to the learner's action, as usual in the bandit case?
- What is the role of Lemma 8 in Appendix B.3 when it is already explained at the end of Section 4.1?

**Audience:**

No

**Audience Explanation:**

I think the way the results in the paper are formalized are too simplistic. With the changes I recommended, they might become interesting to some individuals (although the results are certainly not very exciting in my opinion).

**Broader Impact Concerns:**

None.

**Claims And Evidence:**

No

**Claims Explanation:**

See above.

**Requested Changes:**

See above.

---

### Note · Authors · 2026-02-26

I have read and agree with the venue's withdrawal policy on behalf of myself and my co-authors.